# Target Detection of Diamond Nanostructures Based on Improved YOLOv8 Modeling

**DOI:** 10.3390/nano14131115

**Published:** 2024-06-28

**Authors:** Fengxiang Guo, Xinyun Guo, Lei Guo, Yibao Wang, Qinhang Wang, Shousheng Liu, Mei Zhang, Lili Zhang, Zhigang Gai

**Affiliations:** 1Institute of Oceanographic Instrumentation, Qilu University of Technology (Shandong Academy of Sciences), Qingdao 250316, China; 15315406813@163.com (X.G.); 19986441270@163.com (L.G.); 15102916819@163.com (Y.W.); w1300642955@163.com (Q.W.); liu_ss008@163.com (S.L.); zllouc@163.com (L.Z.); 2National Engineering and Technological Research Center of Marine Monitoring Equipment, Shandong Provincial Key Laboratory of Ocean Environment Monitoring Technology, Institute of Oceanographic Instrumentation, Qilu University of Technology (Shandong Academy of Sciences), Qingdao 250316, China; zhangm2012@126.com; 3Laoshan Laboratory, Qingdao 250316, China

**Keywords:** YOLOv8, DCN_C2f, shuffle attention, WIoU, diamond nanostructure

## Abstract

Boron-doped diamond thin films exhibit extensive applications in chemical sensing, in which the performance could be further enhanced by nano-structuring of the surfaces. In order to discover the relationship between diamond nanostructures and properties, this paper is dedicated to deep learning target detection methods. However, great challenges, such as noise, unclear target boundaries, and mutual occlusion between targets, are inevitable during the target detection of nanostructures. To tackle these challenges, DWS-YOLOv8 (DCN + WIoU + SA + YOLOv8n) is introduced to optimize the YOLOv8n model for the detection of diamond nanostructures. A deformable convolutional C2f (DCN_C2f) module is integrated into the backbone network, as is a shuffling attention (SA) mechanism, for adaptively tuning the perceptual field of the network and reducing the effect of noise. Finally, Wise-IoU (WIoU)v3 is utilized as a bounding box regression loss to enhance the model’s ability to localize diamond nanostructures. Compared to YOLOv8n, a 9.4% higher detection accuracy is achieved for the present model with reduced computational complexity. Additionally, the enhancement of precision (P), recall (R), mAP@0.5, and mAP@0.5:0.95 is demonstrated, which validates the effectiveness of the present DWS-YOLOv8 method. These methods provide effective support for the subsequent understanding and customization of the properties of surface nanostructures.

## 1. Introduction

The relationship between microstructure and performance for electrode materials, such as Boron-doped diamond (BDD), has been a foundational question in instrument science and sensor technology. BDD films have been widely used in electrochemical sensing because of their excellent mechanochemical stability, wide electrochemical windows, low background currents, and excellent resistance to biological fouling [1]. Recent studies have shown that the surface structure and morphology of BDD film electrodes are the key factors in determining their macroscopic sensing properties, such as seawater salinity. Hence, nano-structuring of electrode materials has drawn extensive attention from scientific and technological points of view. However, previous researchers have mainly interpreted the morphology and microstructure by analyzing the SEM photographs, which were strongly determined by the subjective judgment of analysts. With the development of deep learning, the target detection and feature extraction of large-area and complex surface structures could be easily carried out under more objective data-based evaluation criteria [2]. Therefore, target detection by deep learning, as the foundation and bridge for understanding, optimization, and application of diamond nanostructures, plays an important role in promoting automatic analysis and subsequent parameter acquisition and is thus of great research significance.

The effectiveness of deep learning is verified in various fields such as computer vision, speech recognition, and natural language processing [3]. Further, deep convolutional neural networks (DCNNs) push the performance of computer vision systems to a higher level, including image classification [4] and object detection.

For the target detection of nanostructures by deep learning, there are several technological difficulties to be overcome. Firstly, diamond nanostructures need to be photographed using a specific scanning electron microscope due to their small size, in which grayscale images are always obtained with unclear object boundaries. Secondly, during the fabrication process of physical and chemical etching, the nanostructures occlude each other owing to their high number density and high ratio of length and diameter. Thirdly, the noisy data at an oblique angle increases the task and difficulty of target detection.

Deep learning algorithms could overcome all these obstacles, as well as images noise or structure defect, after training with datasets [5]. The majority of current target detection algorithms can be classified into two distinct categories: two-stage and one-stage methods. Two-stage methods, exemplified by the RCNN (regions with CNN features) family [6,7], involve the extraction of candidate frames followed by classification and non-maximal suppression, which is used to refine the prediction. While two-stage detectors exhibit certain advantages in terms of detection accuracy, they also face challenges in terms of training, detection speed, and optimization. In contrast, single-stage methods, exemplified by the SSD (single-shot multi-box detector) [8,9] and the YOLO (you only look once) [10,11] series, are becoming increasingly prevalent, particularly in real-time object detection. Following the proposal of YOLOv1 by Redmon [12] to address the computational complexity of two-stage algorithms, subsequent researchers have developed YOLOv2, YOLOv3, YOLOv4 [13], YOLOv5 [14], YOLOv7 [15], and YOLOX [16], achieving promising results. It must be mentioned that targets in theses natural scenes are multi-scale with prominent color features.

However, diamond nanostructures in SEM images lack color features and complete information due to occlusion. Therefore, more mechanisms should be incorporated to the conventional detection algorithms to accurately detect diamond nanostructure in SEM images.

Recent research work in the field of target detection in SEM images of micro-structures, provide important theoretical guidance and algorithm optimization for the detection of diamond nanostructures. Okuyama Y et al. proposed an automated analysis method for SEM images of semiconductor device cross-sections using deep learning algorithms [17]. This method comprises two image recognition tasks: target detection for determining the coordinates of each cell of the pattern and semantic segmentation for obtaining each region (mask, substrate, and background) boundaries. The combined results of these two tasks, typically feature lengths, such as width and depth, can be measured accurately and immediately. And the extraction speed is 240 times faster than manual measurement. However, the proposed algorithm is only for target detection of individual semiconductor devices and not applicable to the detection of targets occluded from each other. Dengiz et al. used a fuzzy logic and neural network approach to automatically detect grain boundaries in microstructure images (obtained by optical microscopy of noisy data) of high-temperature alloy steels during the sintering process [18]. Albuquerque et al. [19] efficiently segmented and quantified microstructures in nodular, gray, and malleable cast iron images based on multilayer perceptive neural networks. Mulewicz et al. [20] classified different types of microstructural images of metals obtained by optical microscopy techniques based on deep convolutional neural network (DCNN) image analysis techniques. Adachi et al. [21] used three typical convolutional neural networks, including LeNet5, AlexNet, and GoogleNet, to recognize the microstructure of steel, verify their recognition accuracy, and investigate the effect of learning rate, dropout rate, and average image subtraction on the recognition accuracy. The effectiveness of deep learning to classify microstructures was demonstrated. However, all of these algorithms were based on samples taken at a flat overhead angle and do not apply to SEM images at an oblique angle. For most of the target detection methods, for SEM images of common materials, there is a general problem of low accuracy. Yet, there is still no precedent of applying deep learning to target the detection of diamond nanostructures with low image quality.

In order to solve the above problems, a target detection model is proposed based on SEM images of diamond nanostructures, called DWS-YOLOv8, using YOLOv8n as the backbone network. By replacing the backbone network with fused multiscale feature information, adding an attention mechanism and loss function that can reduce noise to improve the target localization ability, the performance of target detection for diamond nanostructures can be enhanced.

## 2. Methods

Taking account of the facts, there is no public dataset of diamond nanostructures. This paper starts by preparing diamond nanostructures and gradually builds the dataset used in this research. Then, the DWS-YOLOv8 model proposed in this paper is optimized on the basis of the original YOLOv8n model in three aspects, namely, network structure, attention mechanism, and loss function, respectively.

### 2.1. Producing Diamond Nanostructure Datasets

#### Materials and Experimental Methods

The BDD films (10 μm) were prepared using a hot filament chemical vapor deposition system (HFCVD) [22]. The deposition conditions were as follows: chamber pressure was 3–5 kPa, hot wire temperature was 2400–2500 °C, base temperature was 850 °C, doping concentration was 8000 ppm (ppm B/C in gas phase), and growth rate was 2.5 μm/h. Combined with a porous anodized aluminum oxide template, a hard mask layer of 200 nm was prepared by e-beam evaporation of 4N (99.99%) nickel metal pillars. Self-organized Ni nanoparticles were obtained after a 4-h vacuum annealing under 700 °C. Next, inductively coupled plasma (ICP) etching was performed to etch the BDD layer for 60 min [23] using the RF plasma equipment ICP601 (Beijing Chuangshiweina Technology, Beijing, China). Plasma was ignited in oxygen and argon atmospheres at pressures of 3 Pa. In the case of O_2_ + Ar gas mixtures, the gases were introduced into the chamber in a 10:1 ratio at a pressure of 3 Pa. The RF power of 600 W was used to stimulate the plasma. After the plasma structuring process, the experimental results revealed that the diamond layer was covered by a re-deposited Ni nanoparticles layer, which served as an etching mask. The as-etched samples were analyzed by secondary electron microscopy (SEM, Hitachi 8100, 45° angle view). A dataset for improved target detection algorithms was constructed using photographs obtained by scanning electron microscopy.

### 2.2. Improvements to the YOLOv8 Model

The YOLO model has had tremendous success in computer vision. Building on this, researchers have improved the method by introducing new modules, leading to the development of many classic models. Compared to the outstanding models of the previous YOLO series, such as YOLOv5 and YOLOv7, YOLOv8 is characterized by higher detection accuracy and speed [24]. The core of the YOLOv8 detection algorithm is the combination of feature extraction, feature fusion, or various feature processing methods to enhance its object detection capability.

#### 2.2.1. Deformable Convolutional DCN_C2f Network Architecture

The design of the YOLOv8 backbone network aims to effectively extract and fuse multiscale feature information. With the development of network architectures, classical backbone networks, such as ResNet, EfficientNet [25], MobileNet [26] and ShuffleNet [27], have been used in YOLOv8 to extract richer feature representations for tasks such as image classification, object detection, and segmentation.

In the backbone network of YOLOv8, the C2f module is responsible for feature extraction from input images [28]. A deep residual network structure is employed, and deep convolutional neural networks (CNNs) are used to extract feature representations. In this module, combined with stacking multiple convolutional layers, features in the image are extracted by utilizing residual connections and skip connections to fully exploit feature information at different levels, thereby enhancing the multiscale representability of features. However, the stacking of this module leads to excessive redundancy in the channel information. In addition, owing to the receptive field of the network restricted by the fixed characteristics of the standard convolutional kernels used in the C2f module, only local object information is captured [29]. As a result, missed detections are common when dealing with multiscale, multitarget, or occluded diamond nanostructures in SEM images. Then, for small target detection tasks such as diamond nanostructures, the C2f module is overly large, computationally intensive, and lacks lightweight characteristics.

To overcome these challenges, experiments are conducted to improve the detection accuracy. The second, third, and fourth C2f modules are replaced in the backbone network of the original YOLOv8 with the DCN_C2f modules. The deformable convolution module (DCN_C2f) is capable of expanding the sensory field of the feature map and concentrating the region of interest, which make the features extracted from diamond-nanostructured targets more discriminative. As applied in the remote sensing images [30], the effectiveness of DCN_C2f in enhancing the target localization ability of the algorithm was indicated, which is believed to effectively and accurately detect the nanostructures in the SEM images. Through this improvement, the adaptive adjustment of the receptive field of the network is achieved when model sampling allows, which effectively increases the sensitivity of the network to irregular shapes and improves the detection performance for targets of different scales and irregular shapes.

An offset for each sampling point is introduced into deformable convolution in the convolution kernel, which allows adaptive sampling beyond the constraints of a regular grid. The structure of the deformable convolution layer is shown in Figure 1. And a comparison between deformable and standard convolution is shown in Figure 2. The deformable convolution layer in the model has two tasks: first, to use convolution to obtain offset values from the input feature map, and second, to derive the output feature map based on the input feature map and the obtained offsets [31]. The representation of the deformable convolution is given by:(1)$$y\left(p_{0}\right)=\sum_{p_{n}\in \Omega }w\left(p_{n}\right)x\left(p_{0}+p_{n}+\Delta p_{n}\right)$$

In Equation (1), $p_{0}$ represents the pixel position in the feature map, and Ω corresponds to the regular grid of the convolution kernel, illustrated here with a 3 × 3 convolution kernel as an example:(2)Ω={(−1,−1),(−1,0),⋯,(0,1),(1,1)}

$p_{n}$ is the position in *Ω*, $\Delta p_{n}$ represents the offset $\left(\left\{\Delta p_{n}=H\times W\times 2N|n=1,\ldots N\right\}\right)$, where N=|Ω|, and 2 denotes the two-dimensional offset (∆x,∆y). Here, x, y, and W denote the input feature map, the output feature map, and the sampling position weight, respectively. However, since the offset $\Delta p_{n}$ is usually fractional, the pixel value of the input feature map x cannot be obtained directly. In general, it is obtained by bilinear interpolation, expressed as follows:(3)$$x\left(p\right)=\sum_{q}G\left(q,p\right)\cdot x\left(q\right)$$

In the equation, p represents an arbitrary $\left(p=p_{0}+p_{n}+\Delta p_{n}\right)$, q denotes the spatial position in the feature map x, and G(·,·) is the bilinear interpolation kernel, expressed as follows:(4)$$G\left(q,p\right)=g\left(q_{x},p_{x}\right)\cdot g\left(q_{y},p_{y}\right)$$
where g(q,p)=max(0.1−|q−p|).

#### 2.2.2. Shuffle Attention Mechanism

Attentional mechanisms are techniques that mimic the process of allocating human attention by allowing a model to focus on the parts of the input data that are relevant to the task at hand while ignoring the rest while processing it. Attention mechanisms are mainly categorized into three domains based on their application differences: spatial domain, channel domain, and mixed domain [32]. In computer vision, attention mechanisms are used in tasks such as image classification, target detection, and image generation. The attention mechanism allows the model to allocate different attention to different regions of the image during classification, thus improving the accuracy of classification [33].

Due to the limited contrast between background and target regions in images acquired by SEM, coupled with the presence of various types of noise, the application of many detection models is restricted by poor background suppression and noise resistance capabilities. To improve the detection model’s focus on the critical information in the input features while minimizing attention to the background, an attention mechanism called shuffle attention was introduced into the model’s backbone network. As it is applied to coal mining machine hobnail teeth [34], the ability to localize the hobnail teeth in dark and noisy environments is improved by the introduction of shuffle attention. And the noise suppression ability of shuffle attention is also proven, which would reduce the noise impact on the diamond nanostructure target detection results.

In the shuffle attention structure, the shuffle unit is used to integrate spatial and channel attention mechanisms. The structure diagram of the shuffle attention mechanism is shown in Figure 3.

For a given diamond nanostructure feature map $X\in R^{C\times H\times W}$, where C, H, and W represent the number of channels, spatial height, and width, respectively. SA first divides X along the channel dimension into G groups, i.e., $X=[X_{1},\ldots ,X_{G}]$, where $X_{k}\in R^{C/G\times H\times W}$. Each sub-feature $X_{k}$ gradually captures specific semantic responses during the training process. Then, the attention module generates importance coefficients for each sub-feature. Specifically, at the beginning of each attention unit, the input of $X_{k}$ is split into two branches along the channel dimension, i.e., $X_{k1}{,X}_{k2}\in R^{C/2G\times H\times W}$. In one branch the inter-channel relationships are used to output channel attention maps, while in the other branch, the spatial relationships of features are used to generate spatial attention maps.

Channel attention first generates channel-wise statistics through global mean pooling, i.e., $s\in R^{C/2G\times 1\times 1}$, which embeds global information. These can be computed by contracting $X_{k1}$ along the spatial dimensions H×W:(5)$$s=F_{gp}\left(X_{k1}\right)=\frac{1}{H\times W}\sum_{i=1}^{H}\sum_{j=1}^{w}X_{k1}\left(i,j\right)$$
where *s* represents the global information of the channel attention, H and W denote the spatial height and width, respectively. And $X_{k1}\left(i,j\right)$ denotes the value of the pixel located at the *i*-th row and *j*-th column of the feature map partitioned by the channel attention.

Additionally, a compact feature is created to guide the precise and adaptive selection. The final output of the channel attention is as follows:(6)$$X_{k1}^{\prime }=\sigma \left(Fc\left(s\right)\right)\cdot X_{k1}=\sigma \left(W_{1s}+b_{1}\right)\cdot X_{k1}$$

Here, $W_{1}\in R^{C/2G\times 1\times 1}$, $b_{1}{\in R}^{C/2G\times 1\times 1}$ are the parameters used to scale and shift s, and σ refers to the sigmoid function, that maps real valued inputs to values in the range between 0 and 1. Its mathematical expression is:(7)$$\sigma \left(x\right)=\frac{1}{1+e^{-x}}$$

In contrast to channel attention, spatial attention focuses on “where” information, providing a complement to channel attention. First, spatial statistics are obtained using the group norm on $X_{k2}$. Then, $F_{c}\left(\cdot \right)$ is used to improve the feature representation. The final output of spatial attention is as follows:(8)$$X_{k2}^{\prime }=\sigma \left(W_{2}\cdot GN\left(X_{k2}\right)+b_{2}\right)\cdot X_{k2}$$

Here, $W_{2}$ and $b_{2}$ are parameters with a shape of $R^{C/2G\times 1\times 1}$.

Group norm is a normalization method that divides channels into multiple groups, with channels within each group sharing statistical information. the mean and variance for the channels within each group is calculated by group norm, and then these statistics are used to normalize the channels within the group. The mathematical expression for group norm is as follows:(9)$$GN\left(x\right)=\gamma \left(\frac{x-\mu }{\sqrt{\sigma^{2}+\epsilon }}\right)+\beta$$
where x is the input tensor, $\mu =\frac{1}{H\times W}\sum_{i=1}^{H}\sum_{j=1}^{w}x_{i,j}^{g}$ represents the mean of channels within each group, and $\sigma^{2}=\frac{1}{H\times W}\sum_{i=1}^{H}\sum_{j=1}^{w}{{(x}_{i,j}^{g}-\mu )}^{2}$ denotes the variance of channels within each group, $x^{g}$ denotes the set of channels within the group. The scaling coefficient γ and the offset coefficient β, obtained during training, are applied. The constant ϵ is introduced to prevent division by zero errors in the division operation. H and W, respectively, represent the height and width of the input tensor, and $x_{i,j}^{g}$ denotes the channel value at position (i,j) within the g-th group.

Finally, all the channel attention and spatial attention obtained from the diamond nanostructure images are aggregated:(10)$$X_{k}^{\prime }=\left[X_{k1}^{\prime },X_{k2}^{\prime }\right]\in R^{\frac{C}{2G}\times H\times W}$$

At this point, the output feature map has the same dimensions as the input.

In this paper, the SA attention mechanism is incorporated into the model. On the one hand, shuffle attention helps to reduce the model’s attention to background information. On the other hand, the mixed-domain attention mechanism enhances the model’s focus on crucial target information, thereby optimizing the model’s detection performance.

#### 2.2.3. WISE-IoUv3 Loss Function

In target detection tasks during SEM image scenarios with unclear target boundaries, mutually occluded objects, and poor image quality, the model’s detection performance can be significantly improved by designing an appropriate loss function. For the purpose of optimizing the loss function, the Wise-IoU (WIoU) [35] method was utilized. The application of the WIoUv3 loss function to the detection model of small targets [36] improved the model’s focus on small samples in UAV aerial photography scenarios. Thus, it is believed that the detection ability, accuracy, and stability of the model in detecting diamond nanostructure targets of the WIoUv3 loss function would be improved.

Wise-IoU introduces category weighting based on traditional IoU to minimize differences between categories and reduce their impact on test results [37]. This involves assigning a weight to each category and using different weights to weigh the overlap between different categories when calculating the IoU, thereby achieving more accurate evaluation results. Among the three versions of WIoU, the focusing coefficients were introduced in WIoUv3.

The parameter scheme of Wise-IoU is shown in Figure 4. A dynamic non-monotonic focusing mechanism is used to evaluate the anchor box quality of diamond nanostructures. The use of gradient amplification reduces the influence of harmful gradients ensuring the high-quality anchor box effect of diamond nanostructures, thereby improving the overall performance of the algorithm.

Wise-IoUv3, building upon the foundation of Wise-IoUv1, introduces a non-monotonic focusing coefficient ‘*r*’ constructed through the outlier degree parameter β. The formula is as follows:(11)$$L_{WIoUv3}=rL_{WIoUv1}$$
(12)$$L_{WIoUv1}=R_{WIoU}L_{IoU}$$
(13)$$R_{WIoU}=\exp\left(\frac{{\left(x-x_{gt}\right)}^{2}+{\left(y-y_{gt}\right)}^{2}}{{\left(W_{g}^{2}+H_{g}^{2}\right)}^{*}}\right)$$
(14)$$r=\frac{\beta }{\delta \alpha^{\beta -\delta }}$$
(15)$$\beta =\frac{L_{IOU}^{*}}{\bar{L_{IOU}}}$$

In the equation, the superscript ‘*’ denotes the separation of $W_{g}$ and $H_{g}$ from the computational graph, which effectively improves convergence efficiency. β is defined as the outlier degree, where a smaller outlier degree implies a higher quality anchor box. It is assigned a small gradient gain to focus the bounding box regression on anchor boxes of normal quality. Anchor boxes with a larger outlier degree are assigned a small gradient gain to effectively prevent low quality data from generating significant detrimental gradients. α and δ are hyperparameters, and when β=δ, r=1. If the outlier degree of an anchor box satisfies β=C (where C is a constant), the anchor box receives the highest gradient gain. Since both $L_{IOU}$ and the quality criteria for anchor boxes are dynamic, WIOUv3 can dynamically assign gradient gain strategies that best fit the current situation at any time.

$\bar{L_{IOU}}$ is the moving average with a momentum of m, which is dynamically updated to maintain an overall high level of β. This effectively addresses the issue of slow convergence speed in the later stages of training.

In this paper, WIoUv3 is introduced in the regression loss. On the one hand, the excellent loss function is adhered in WIoUv3. On the other hand, during the process of target detection of diamond nanostructures, occluded samples have less impact in traditional training, leading to underfitting of the model. With the dynamic gradient gain strategy, these samples are assigned higher gradient weights, and more consideration is given to these occluded scenes when the model parameters are updated, which enhances the detection ability of the occluded scenes, balances the influence of different loss terms, and accelerates the model convergence. These reduce the risk of overfitting, which enables the model to better focus on and optimize the occluded samples and key features, thus improving the overall detection accuracy. For the mutual occlusion during target detection tasks in SEM scenes, the loss weights for targets are dynamically optimized to improve the model’s detection performance.

The finally improved network model is called DWS-YOLOv8, and the overall framework of the model is shown in Figure 5.

The schematic workflow diagram illustrating the training process of DWS-YOLOv8 is shown in Figure 6. As it is shown, the training process of DWS-YOLOv8 includes data preparation, model initialization, forward propagation, loss computation, backpropagation, optimizer update, training iteration, model validation, model saving, and test evaluation.

## 3. Experiments and Results Analysis

### 3.1. Datasets

The original image, captured by a scanning electron microscope, was divided into six smaller images of equal dimensions. These images were then integrated to create a dataset of diamond nanostructures, designated as the NanoData dataset. The original image captured by the scanning electron microscope is depicted in Figure 7. A portion of the diamond nanostructures is present in the NanoData dataset. as illustrated in Figure 8.

The NanoData dataset utilized in this study comprises two annotated object classes: nanopillars and nanocones. As depicted in Figure 9, the pertinent information pertaining to the manually annotated objects within this dataset is illustrated. The subfigures in Figure 8 are presented in order from left to right and top to bottom. Figure 9a depicts the number of objects in each category in the dataset. Figure 9b depicts the size of the object enclosure boxes in the dataset, with the center coordinates of all the object enclosure boxes fixed at a single point. Figure 9c illustrates the distribution of the center coordinates of the object enclosure boxes. Figure 9d illustrates a scatter plot of the corresponding widths and heights of the object enclosure boxes.

### 3.2. Evaluation Metrics

To quantitatively analyze the training results of target detection, researchers have established various evaluation metrics, each designed to reflect a specific aspect of the performance of the detection algorithm. In this paper, the mean average precision (mAP), precision (*P*), recall (*R*), are primarily included in the precision-related metrics [38].

(1) Precision

Precision (*P*) is the proportion of all samples predicted as true by the model that are, in fact, true. It is calculated using the following formula:(16)$$P=\frac{TP}{TP+FP}$$

*TP* represents the number of instances in which the classifier predicts positive samples correctly, that is, the actual positive samples that are correctly identified. Conversely, *FP* signifies the instances in which the classifier predicts positive samples, but they are, in fact, negative samples. This parameter indicates the number of falsely reported negative samples.

(2) Recall

The recall (*R*) is the proportion of all actual true samples that are correctly identified by the model. It is calculated using the following formula:(17)$$R=\frac{TP}{TP+FN}$$
where *TP* represents true positives, and *FN* represents false negatives.

(3) mAP

The metric average precision (*AP*) is utilized in the context of target detection to assess the performance of a model in detecting targets. The calculation formula is as follows:(18)$$AP=\sum_{n}\left(R_{n}-R_{n-1}\right)\cdot P_{n}$$
where $R_{n}$ represents the recall at the *n*th precision point, $P_{n}$ represents the precision at the *n*-th recall point, and the sum is taken over all precision–recall pairs.

The mean average precision (*mAP*) is the average of the average precision (*AP*) values across all classes. The calculation formula is as follows:(19)$$mAP=\frac{\sum_{0}^{N}AP_{n}}{N}$$
where N represents the total number of classes, $AP_{n}$ denotes the average precision for the nth class, corresponding to the area under the precision–recall curve. 

mAP@0.5 signifies the mean average precision value when the IoU parameter is set at a threshold of 0.5. The average value of mAP is represented as mAP@0.5:0.95, indicating the range of IoU parameter thresholds as [0.5:0.05:0.95].

### 3.3. Experimental Environment and Hyperparameter Settings

The experimental environment is shown in Table 1.

In order to meet the needs of different scenarios and applications to balance the speed, accuracy, and resource consumption of the model, the YOLOv8 model generates five different model versions by adjusting the depth and width of the network structure; they are YOLOv8n, YOLOv8s, YOLOv8m, YOLOv8l, and YOLOv8x. The parameters and computation amount of the five models increase sequentially, and the accuracy of the detection increases gradually. The corresponding channel width, depth and maximum number of channels of these five models are shown in Table 2.

In order to better adapt to the training environment, as well as better improve the model, YOLOv8n was chosen as the baseline model in this research, and the hyperparameter settings in the model training are shown in Table 3.

The training process incorporates a learning rate decay method, whereby the initial learning rate (learning rate 0, lr0) controls the rate at which the model parameters are updated. The coefficient of the initial learning rate (learning rate float, lrf) adjusts the decay of the learning rate during the training process to obtain the final learning rate, which is obtained by multiplying the initial learning rate with the coefficients. An epoch represents a complete iteration of the entire image dataset in YOLO. During each epoch, the YOLO model traverses all the bounding boxes in the dataset and updates its model parameters according to the aforementioned loss function and optimization algorithm. The higher the number of epochs, the more times the YOLO model traverses the entire dataset. To ensure the stability of the training process, the training was performed for 180 rounds, during which the learning rate gradually decreased. This configuration permits the model to converge smoothly to the optimal solution, thereby avoiding oscillations.

### 3.4. Results

On the basis of the NanoData dataset, the present model was compared with the YOLOv8 model, the common excellent target detection models, and the YOLOv8 model equipped with a single improved strategy in the comparison experiments, respectively, and a visualization analysis was carried out, which was used to illustrate the superiority of the present model for the target detection of diamond nanostructures.

#### 3.4.1. Comparison with YOLOv8

In order to evaluate the improvement of the enhanced model in terms of detection accuracy, this paper conducts comparative experiments between the YOLOv8 model and the DWS-YOLOv8 model using both the training and validation sets. Figure 10 illustrates the changes in several key evaluation metrics throughout the training process for DWS-YOLOv8 and YOLOv8n. Precision, recall, and mAP@0.5 improve rapidly throughout the iterations, gradually approaching stable values. Figure 11 shows the regression loss of DWS-YOLOv8 and YOLOv8 during training and testing. Table 4 demonstrates the computational complexity and resource requirements of the DWS-YOLOv8 model compared to the YOLOv8 base model.

As illustrated in Figure 10, DWS-YOLOv8 outperforms YOLOv8n in the three detection metrics of precision, recall, and mAP0.5 after approximately 80 epochs of training. It is demonstrated that DWS-YOLOv8 is able to recognize targets more accurately and has higher detection performance in the diamond nanostructures target detection task. In the early stages of model training, it is normal for the model to show signs of overfitting or underfitting to the training data, resulting in some variations and fluctuations. DWS-YOLOv8 stabilizes after about 120 epochs, and the performance no longer changes significantly and stabilizes at a high level, proving that the model converges to a better state and no longer needs substantial optimization.

As illustrated in Figure 11, as the number of iterations increases, the average value of the loss function decreases significantly. The loss function of DWS-YOLOv8 accelerates convergence and improves prediction accuracy, and the average value of the loss function tends to converge as the number of trainings approaches 180.

As shown in Table 4, the computational complexity and resource requirements of the model are significantly increased in DWS-YOLOv8 compared to YOLOv8n due to the introduction of DCN_C2f and SA. This is attributed to the fact that deformable convolution requires the computation of additional offsets and weights, and the attention mechanism requires the global weighting computation of the feature map, and these operations increase the number of parameters and the computational effort.

Since YOLOv8n is used as the benchmark model in this model, to further illustrate the effectiveness of this paper’s method, the model in this paper is compared with several YOLOv8s (YOLOv8s, YOLOv8m, YOLOv8l, and YOLOv8x) of different sizes in the dataset. The experimental results are shown in Table 5.

Recall is the ability of the model to correctly identify the target, mAP@0.5 is the average precision of the model at different confidence thresholds, and mAP@0.5:0.95 is the average precision of the model at higher confidence thresholds. As illustrated in Table 5, the enhanced model exhibits the best detection performance across the four evaluation metrics of recall, mAP0.5, mAP0.5:0.95, and F1-score, in comparison to the other models, particularly the model with a larger scale than itself. While this model may exhibit lower detection accuracy compared to larger-scale models, its smaller size results in significantly faster training speeds. Thus, sacrificing a small portion of detection accuracy enables a swift detection process. These results indicate the excellent detection performance of the present model besides keeping a relatively small scale.

#### 3.4.2. Ablation Experiment

In order to evaluate the performance gain of the three strategies on the benchmark model (YOLOv8n), namely the loss function of the DCN_C2f module, the SA module, and the Wise-IoUv3 optimization, the corresponding models are trained and tested on the NanoData dataset. The experimental results are shown in Table 6, which demonstrates the effects of the model combination and improvement on the dataset. As depicted in Table 6, the symbol √ indicates the improved strategy that was used.

The experimental results in Table 6 indicate that each improvement strategy, when applied to the base model, enhances the detection performance to varying degrees. By applying DCN_C2f to YOLOv8n, the network’s receptive field with deformable convolutions is enlarged, and sampling is closer to the objects. Then, the test set precision (P) is increased by 4.3%. Correspondingly, the recall (R), mAP@0.5, and mAP@0.5:0.95 increase by 4.9%, 3.2%, 0.5%, respectively. When SA is applied to YOLOv8n, its efficient attention mechanism enhances the focus on key information in the feature map. These result in an increase of 1.3% in precision (P) and a 2.1% improvement in precision (P) when they are applied to the YOLOv8n model simultaneously with DCN_C2f. When WIoUv3 and a smarter sample allocation strategy are introduced into the regression loss to enhance the localization ability of the model, the precision (P) and recall (R) are improved by 4.2% and 2.9%, respectively. After the three strategies are applied to the YOLOv8n model simultaneously, the model obtains optimal overall performance.

Obviously, DCN_C2f is applied to achieve the optimal values for recall, mAP@0.5, and mAP@0.5:0.95. Considering the impact of attention mechanisms and loss functions on the accuracy of diamond nanostructure detection, a portion of recall and mAP@0.5 is sacrificed to achieve higher detection accuracy with reduced occurrences of missed detections and false alarms.

#### 3.4.3. Comparative Experiment

To demonstrate the superiority and effectiveness of the improvement algorithm proposed in this paper, this paper conducted comparative experiments between DWS-YOLOv8 and other excellent models (such as SSD, Faster R-CNN, CenterNet and YOLOv7). The experimental results are shown in Table 7.

These models selected in this paper are all mainstream algorithms in the field of target detection, representing different stages of development and technical routes of target detection. Among them, SSD and YOLOv7 are single-stage detectors, while Faster R-CNN is a two-stage detector and CenterNet is based on centroid detection. To comprehensively understand the performance of the optimized YOLOv8n models, the accuracy, F1-score, and computational resource consumption are assessed and compared.

The results in the table show that, compared to other mainstream models, the proposed DWS-YOLOv8n in this paper has better detection performance. The mAP0.5 and F1-score of the proposed model in this paper are higher than those of other excellent models, which proves that this model recognizes the correct object more accurately in detecting the target, and the target detection accuracy is higher. This is attributed to the DCN_C2f module used in this paper, which adaptively adjusts the perceptual field of the network and expands the perceptual range of the feature map, thus making the features extracted from diamond-nanostructured targets more discriminative.

As for the value of mAP0.5:0.95, the model proposed in this paper is slightly lower than the SSD model and the Faster R-CNN model. This is because of the two-stage approach used in SSD and Faster R-CNN, which is more fine-grained in generating candidate frames in the first stage, and thus obtains a higher mAP0.5:0.95. However, in this paper, a single-stage target detection model is adopted. At the cost of slightly lower mAP0.5:0.95, excellent detection speed, higher mAP0.5, and a F1-score are obtained for the present single-stage target detection model. Obviously, this model outperforms the SSD model and the Faster R-CNN model in terms of comprehensive detection capability.

#### 3.4.4. Visualization Analysis

In order to visually and conveniently demonstrate the detection performance of DWS-YOLOv8, the confusion matrix, model inference results, and heatmaps are utilized to analyze the model’s detection performance under comparative experiments. 

To visualize the performance of the DWS-YOLOv8 model on the dataset, this research generates confusion matrices for both YOLOv8n and DWS-YOLOv8 to compare the target detection performance of the two models on diamond nanostructures, as shown in Figure 12. The rows and columns of the confusion matrix represent the actual and predicted categories, respectively. There are two categories in the dataset: nanocone and nanopillar. The diagonal values represent the percentage of correct predictions for each category, while the values in other areas indicate the proportion of incorrectly predicted categories.

As shown in Figure 12, it can be seen that the diagonal region of the confusion matrix in DWS-YOLOv8n is darker, indicating that model’s ability to correctly predict object categories has been enhanced. In addition, the values on the diagonal are larger, indicating that the prediction accuracy of this model is higher than that of the YOLOv8n model.

To visually demonstrate the detection performance of this model, this study conducts inference experiments using YOLOv8n and DWS-YOLOv8. This research selects three SEM images containing most of the diamond nanostructures outside the dataset, which include a large number of diamond nanostructures. The visualization results of target detection on the test set for YOLOv8n and the improved YOLOv8 model are shown in Figure 13.

As shown in Figure 13, it can be observed that the improved YOLOv8 model shows significant optimization in detecting multiple targets, occluded targets, etc., and the overall target detection performance. The experimental results confirm that the DWS-YOLOv8 model significantly reduces leakage and misdetection, which promotes to detect diamond nanostructures more accurately. 

In order to visualize the detection performance of the model for different scales of diamond nanostructures, two different scales of diamond nanostructures were selected in this study. The detection results of this model for different scales of diamond nanostructures are depicted in Figure 14.

As shown in Figure 14, it can be seen that the model has good detection abilities for diamond nanostructures with diameters ranging from 80 nm to 500 nm [39]. Generally, the etching process determines the size and shape of the nanostructures. Thus, the dataset constructed accordingly is the most efficient and repeatable for recognizing similar structures. It should be mentioned that, for diamond nanostructures of different scales, the detection accuracy of the model is about 90%. The results show that the model has excellent detection ability for diamond nanostructures of different scales.

Gradient-weighted Class Activation Mapping (Grade CAM) is used to generate heatmaps for YOLOv8n and DWS-YOLOv8. Heatmaps intuitively and conveniently reflect which areas of the feature maps where the model is focusing on. Grade CAM propagates the model’s gradient information backward to the last convolutional layer and then combines it with the weights of the feature maps to generate spatial region features related to the model’s decisions. Pixels with higher gradients in the heatmap are represented by red shading, while the ones with lower gradients are represented by blue shading. The experimental results are shown in Figure 15.

For the model proposed in this paper, the heat map shows that the attention of the model is more focused on the center point of the object, which proves that this model has high accuracy in target localization. This indicates that the model pays more attention to the main part of the target, which can help the model identify and classify the diamond nanostructures better.

## 4. Conclusions and Future Direction

Diamond nanostructures have been successfully prepared at the nanoscale by employing plasma etching techniques. In order to comprehensively analyze and examine the characteristics of these nanostructures, high-quality morphology images were obtained using a high-resolution scanning electron microscope (SEM). These images clearly demonstrate the surface morphology of the nanostructures as well as the periodic arrangement details. Based on the acquired high-precision topographic images, this research further constructed an exhaustive dataset. This dataset systematically organizes and classifies the image information of various nanostructures, which provides valuable data support for subsequent structural design optimization, performance prediction, and an in-depth understanding of the plasma etching process.

When performing target detection of diamond nanostructures in SEM scenarios, problems such as noise, unclear object boundaries, and object occlusion are encountered. The target detection and its accuracy are low for existing models, which makes it difficult to detect fabricated diamond nanostructures. In order to optimize the detection performance of the model, this paper proposes a diamond nanostructure target detection model, DWS-YOLOv8, based on YOLOv8 in SEM scenes. In order to enhance the feature representation for multiscale, occluded diamond nanostructures, adaptively adjusting the network’s receptive field was achieved by the introduction of the lightweight DCN_C2f module. Then, a dynamic weighted attention mechanism was incorporated during feature fusion, which addresses the issue of detail loss during convolutional iterations, facilitates feature self-calibration, to addresses the issue of detail loss, and reduces the impact of noise and background. 

To further improve the training effectiveness and detection accuracy of diamond nanostructures, the dynamic non-monotonic focusing mechanism and gradient gain method were introduced to reduce the influence of harmful gradients. 

Verified by the comparative performance of three improvement strategies, the present DWS-YOLOv8 model achieved improvements in precision (P), recall (R), mAP@0.5, and mAP@0.5:0.95. These demonstrate that the present strategies have enhanced the effectiveness of object detection. Comparative experimental results with other mainstream object detection models, such as Faster R-CNN, SSD, CenterNet, and YOLOv7, also indicate that DWS-YOLOv8n outperforms mainstream models in terms of mAP@0.5 and F1-score. This suggests that the detection accuracy of this model’s diamond nanostructure surpasses those of mainstream models.

Since the improved model added attention mechanisms, the model structure has become more complex, resulting in an overall increase of 0.2 M in model size, and leading to varying degrees of increase in computational and inference time. For some very small diamond nanostructures, the detection accuracy of the improved model is still not high enough. The primary focus of the next step of this research is to further optimize the detection accuracy of the model, with the consideration of resource consumption. In future research, this research will collect more SEM images of multi-diamond nanostructures to serve the subsequent feature extraction of diamond nanostructures.

## Figures and Tables

**Figure 1 nanomaterials-14-01115-f001:**
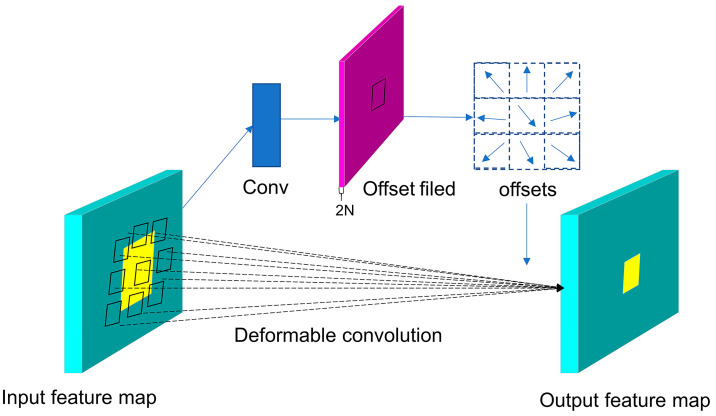
Structure of the deformable convolutional layer.

**Figure 2 nanomaterials-14-01115-f002:**
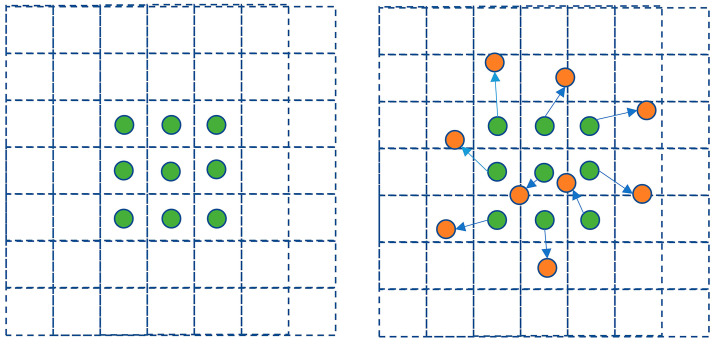
Comparison between deformable convolution and standard convolution.

**Figure 3 nanomaterials-14-01115-f003:**
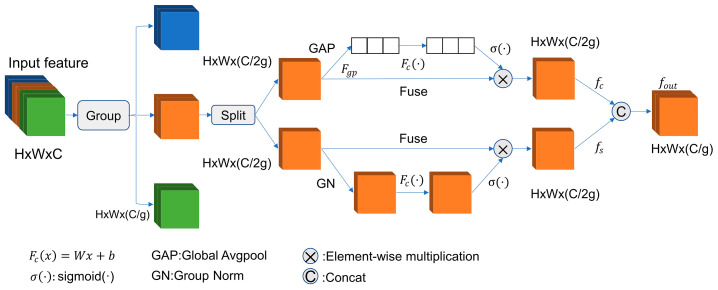
Structural diagram of the shuffle attention mechanism.

**Figure 4 nanomaterials-14-01115-f004:**
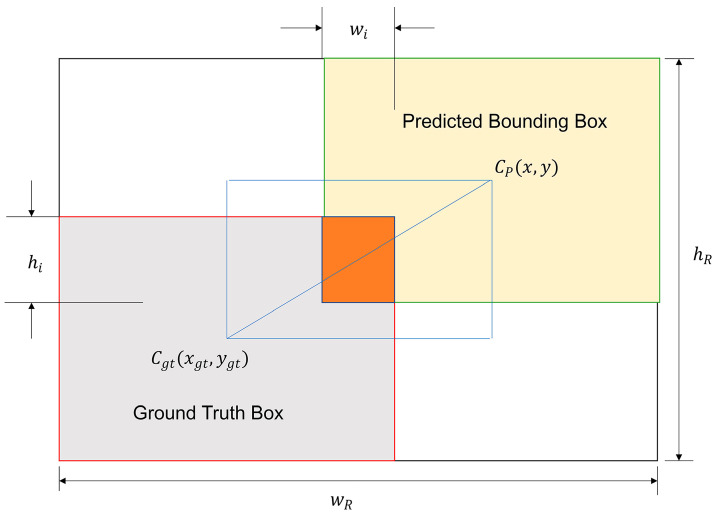
Illustrates the schematic diagram of Wise-IoU parameters.

**Figure 5 nanomaterials-14-01115-f005:**
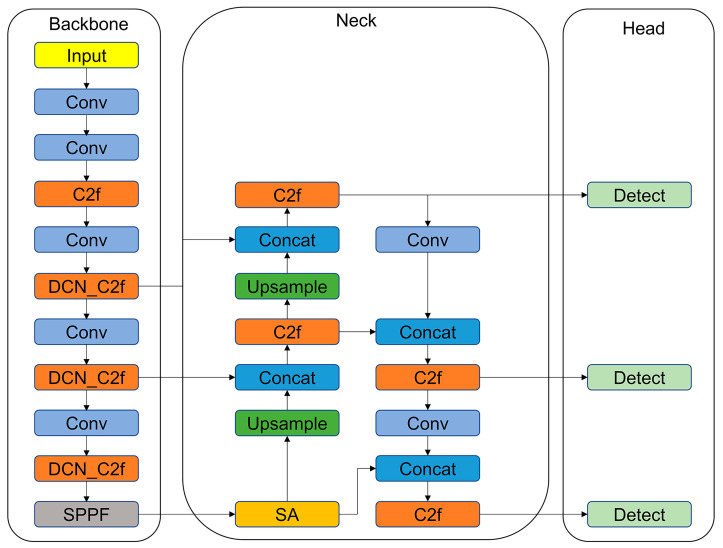
The network structure of the enhanced YOLOv8 is illustrated.

**Figure 6 nanomaterials-14-01115-f006:**
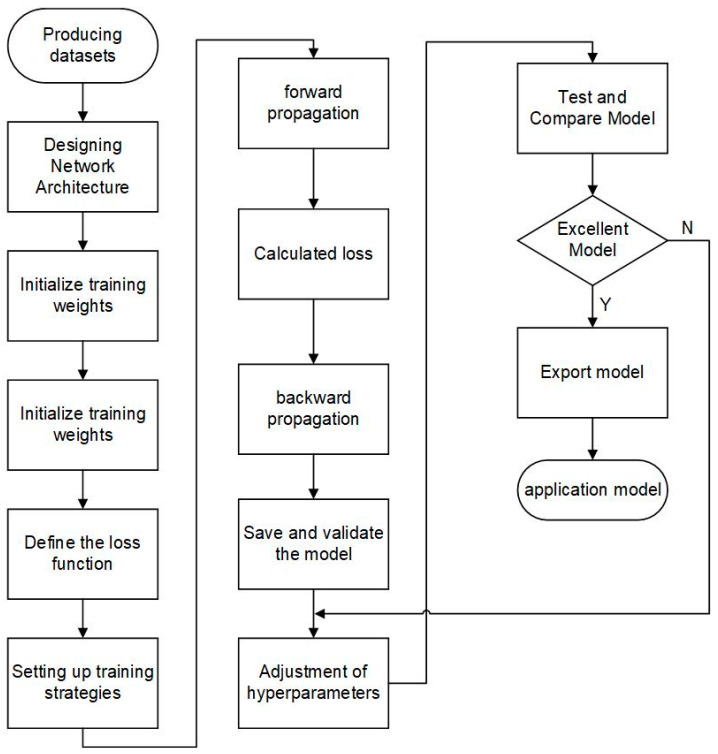
Training process of DWS-YOLOv8.

**Figure 7 nanomaterials-14-01115-f007:**
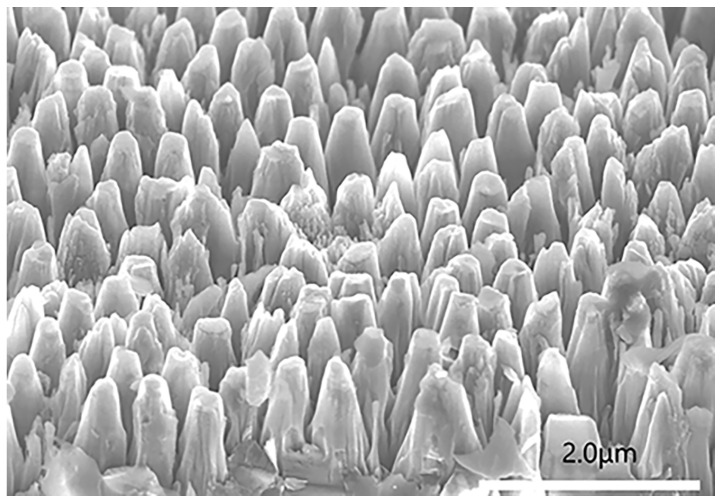
Raw images of diamond nanostructures taken by SEM.

**Figure 8 nanomaterials-14-01115-f008:**
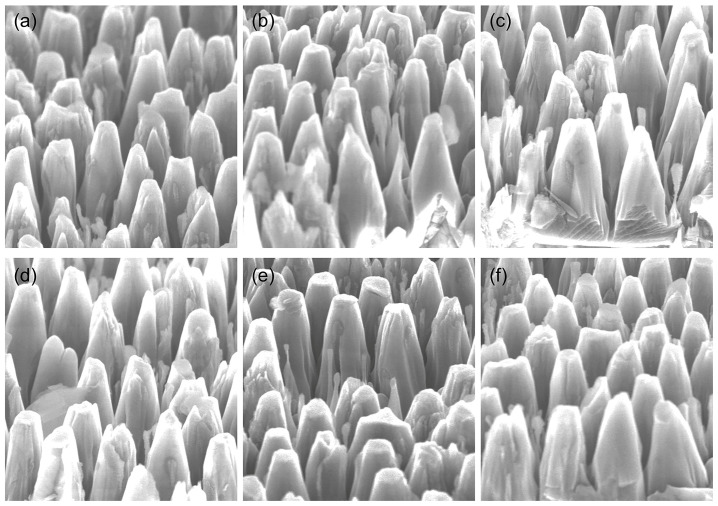
Images of some of the diamond nanostructures in the NanoData. (**a**–**f**) represent images of the diamond nanostructures used in the dataset.

**Figure 9 nanomaterials-14-01115-f009:**
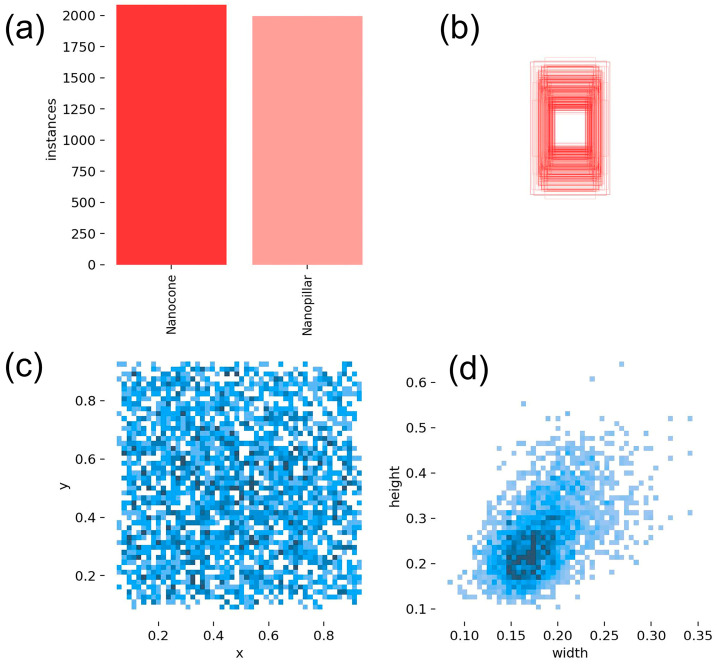
The provided information about manually labeled objects in the NanoData dataset. (**a**) depicts the number of objects in each category in the dataset. (**b**) depicts the size of the object enclosure boxes in the dataset. (**c**) illustrates the distribution of the center coordinates of the object enclosure boxes. (**d**) illustrates a scatter plot of the corresponding widths and heights of the object enclosure boxes.

**Figure 10 nanomaterials-14-01115-f010:**
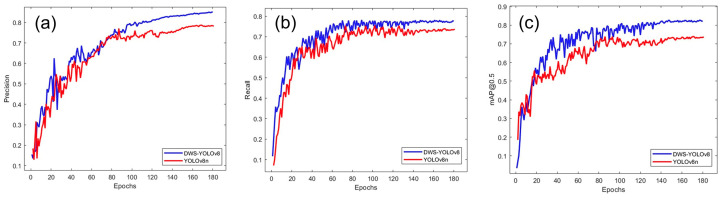
Precision, Recall, and mAP@0.5 for DWS-YOLOv8 and YOLOv8n. (**a**) represents the comparison of accuracy, (**b**) represents the comparison of recall, and (**c**) represents the comparison of mAP@0.5.

**Figure 11 nanomaterials-14-01115-f011:**
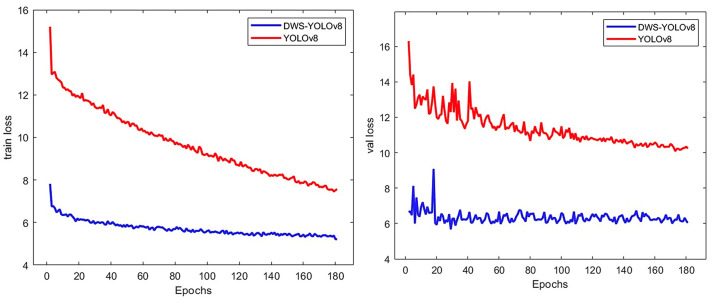
Comparison of training and validation loss curves between DWS-YOLOv8 and YOLOv8n.

**Figure 12 nanomaterials-14-01115-f012:**
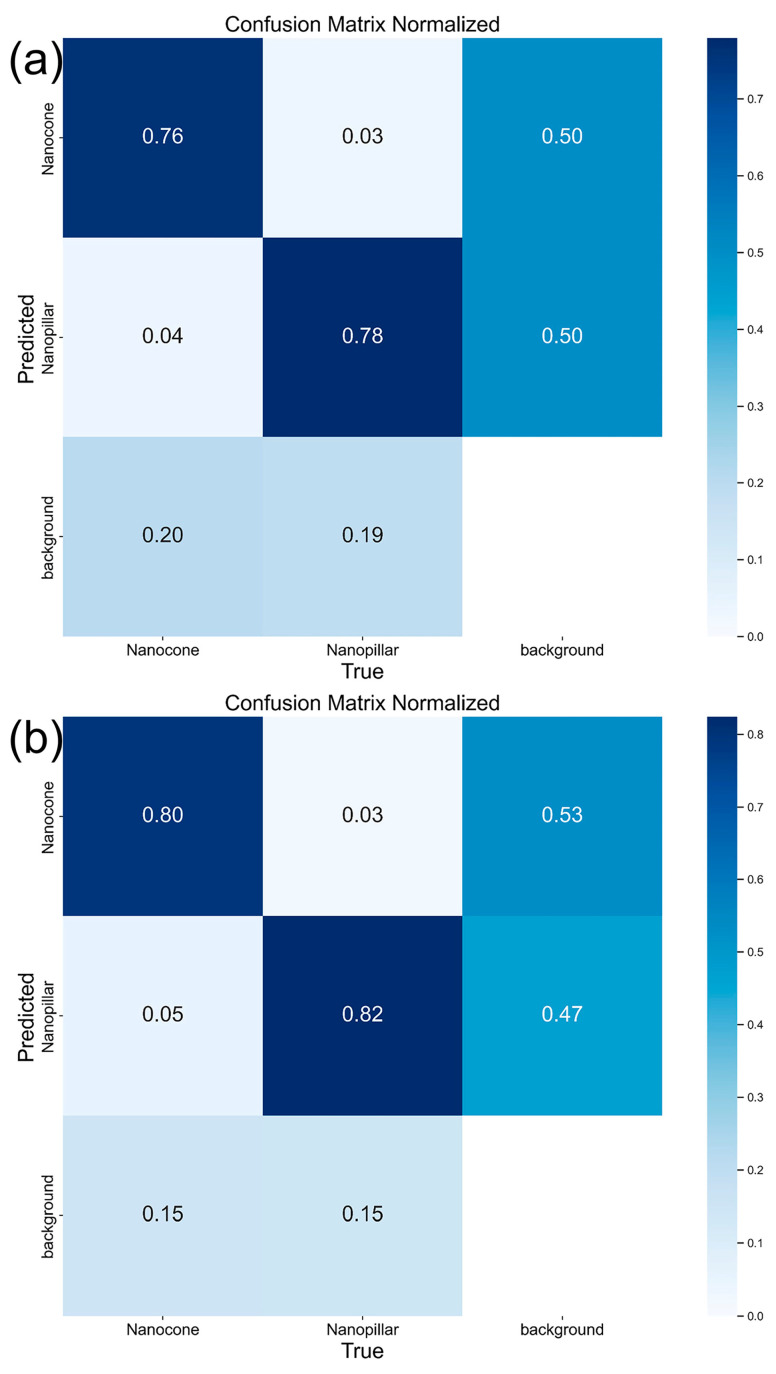
(**a**) Confusion matrix for YOLOv8n; (**b**) Confusion matrix for DWS-YOLOv8.

**Figure 13 nanomaterials-14-01115-f013:**
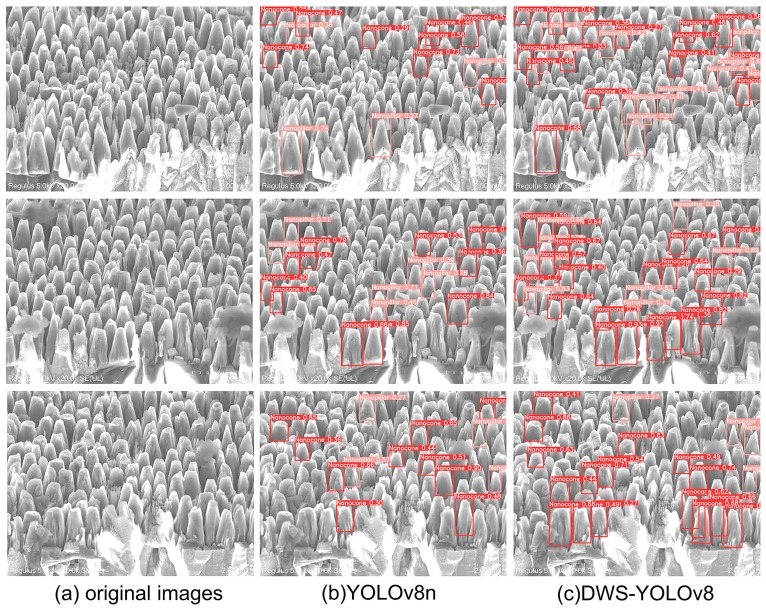
Comparison of target detection results between YOLOV8n and DWS-YOLOv8 on the diamond dataset. (**a**) represents the original image, (**b**) represents the target detection result of YOLOv8n, and (**c**) represents the target detection result of DWS-YOLOv8.

**Figure 14 nanomaterials-14-01115-f014:**
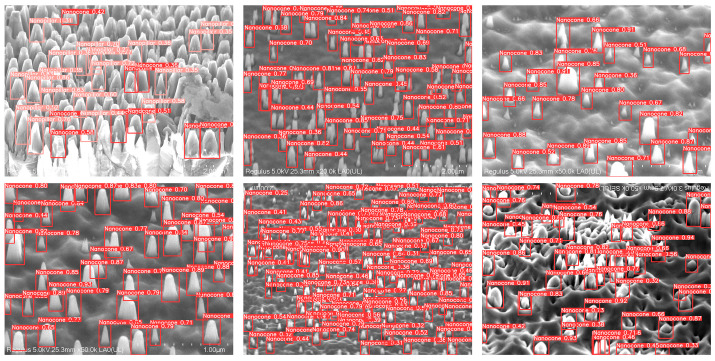
Detection of diamond nanostructures at different scales.

**Figure 15 nanomaterials-14-01115-f015:**
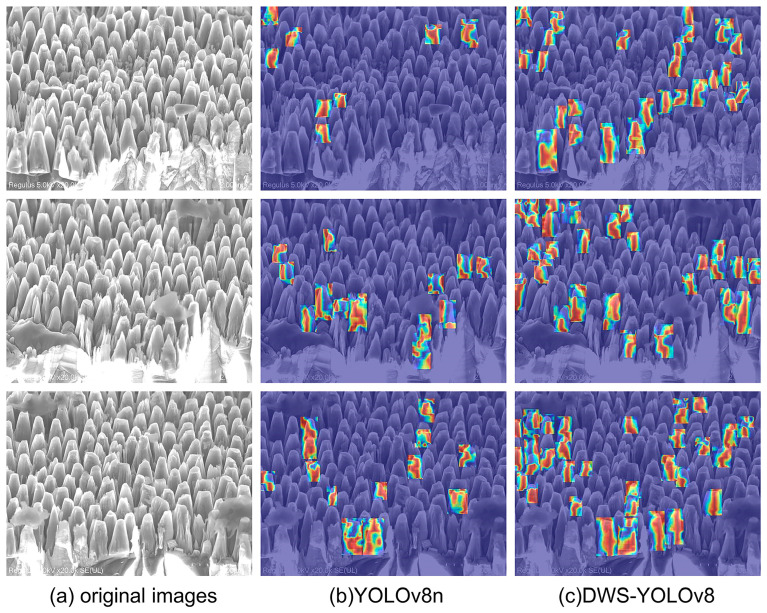
Thermogram comparison of YOLOV8n and DWS-YOLOv8. (**a**) represents the original image, (**b**) shows the thermogram of YOLOv8n, and (**c**) shows the thermogram of DWS-YOLOv8.

**Table 1 nanomaterials-14-01115-t001:** Experimental environment.

Options	Configuration
Operating System	Windows10
CPU	AMD R7 5700X
GPU	NVIDIA RTX3060
GPU memory size	12 G
DL Framework	Pytorch 1.13.1
Language	Python 3.10.11

**Table 2 nanomaterials-14-01115-t002:** Corresponding parameters for different versions of the YOLOv8 model.

Model	Width	Depth	FLOPs	Max Channels
YOLOv8n	0.25	0.33	8.7	1024
YOLOv8s	0.50	0.33	28.6	1024
YOLOv8m	0.75	0.67	78.9	768
YOLOv8l	1.00	1.00	165.2	512
YOLOv8x	1.25	1.00	257.8	512

**Table 3 nanomaterials-14-01115-t003:** Hyperparameter settings.

Hyperparameter Options	Setting
Input Resolution	640 × 640
Initial Learning Rate 0 (lr0)	0.01
Learning Rate Float	0.01
Momentum	0.937
Weight_decay	0.0005
Batch_size	2
Epochs	180

**Table 4 nanomaterials-14-01115-t004:** Computation complexity and resource requirements of DWS-YOLOv8 vs. YOLOv8n.

Models	Model Size/MB	Detection Time/ms	Parameter/10^6^	GFLOPs
YOLOv8n	6.3	8.0	3.0	8.7
DWS-YOLOv8	21.4	15.0	11.1	28.6

**Table 5 nanomaterials-14-01115-t005:** Comparison of the model in this paper with other YOLOv8s of different sizes.

Models	Precision/%	Recall/%	mAP0.5/%	Map0.5:0.95/%	F1-Score
YOLOv8s	78.3	68.0	70.6	54.0	0.72
YOLOv8m	83.1	71.4	74.7	55.6	0.76
YOLOv8l	84.7	73.3	76.9	58.2	0.78
YOLOv8x	85.3	74.9	78.1	60.3	0.79
DWS-YOLOv8	82.4	75.7	81.5	60.7	0.80

**Table 6 nanomaterials-14-01115-t006:** Experimental results of model combinations on the dataset.

Model ^1^				Precision/%	Recall/%	mAP0.5/%	Map0.5:0.95/%
B	D	S	W	DATA	DATA	DATA	DATA
√				73.0	75.1	78.9	60.1
√	√			77.3	80.2	82.1	60.7
√		√		74.3	77.8	80.2	59.4
√			√	77.2	78.0	80.3	59.6
√	√	√		79.4	75.8	81.4	60.5
√	√	√	√	82.4	75.7	81.5	60.7

^1^ B: Base(YOLOv8n); D: DCN_C2f; S: Shaffule attention; W: Wise-IoUv3.

**Table 7 nanomaterials-14-01115-t007:** Comparison with other excellent models.

Models	mAP0.5/%	mAP0.5:0.95/%	F1-Score	GFLOPs
SSD	77.3	61.8	0.53	68.3
Faster R-CNN	79.8	66.2	0.57	932.5
CenterNet	73.3	56.5	0.53	133.7
YOLOv7	80.7	59.6	0.77	105.1
DWS-YOLOv8	81.5	60.7	0.80	79.1

## Data Availability

The derived data generated in this study will be shared by the respective authors upon reasonable request.

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
