# Peer review of "Target Detection of Diamond Nanostructures Based on Improved YOLOv8 Modeling"

_nanomaterials, 2024, doi:10.3390/nano14131115_

Round 1

Reviewer 1 Report

Comments and Suggestions for Authors

Manuscript ID: 3043275

The manuscript, titled " Target detection of diamond nanostructures based on improved YOLOv8 modeling" enhances the detection accuracy of diamond nanostructures in SEM images, crucial for electrochemical sensing electrodes. The study employed the NanoData dataset of SEM images and improved the YOLOv8 model by integrating the DCN_C2f module, SA module, and Wise-IoUv3 optimization strategy. These enhancements were evaluated through extensive experiments focusing on precision, recall, and mean average precision (mAP) metrics. The Wise-IoUv3 optimization demonstrated the most significant improvement, substantially boosting both precision and recall rates. This research shows that the enhanced YOLOv8 model significantly outperforms the baseline in detecting diamond nanostructures, offering valuable advancements for electrochemical sensor technology.

The following suggestions may improve the proposed manuscript:

1.     In the current work, the authors focused on improving a YOLOv8 model. How repeatable do you think this research is?

2.     More experimental nanodiamond samples with various grain sizes should be included to confirm the obtained results.

3.     Further investigation is needed to understand the specific impact of each enhancement strategy (DCN_C2f module, SA module, Wise-IoUv3 optimization) on overall detection performance.

4.     Clarify how the dynamic assignment of gradient gain strategies in Wise-IoUv3 contributes to improved detection accuracy.

5.     Discuss the computational complexity and resource requirements of the enhanced model compared to the base YOLOv8 model.

6.     In the introduction section, there are many typo mistakes such as “previ-ous”, “ana-lyzing”, and “ap-plication”. Please revise the manuscript.

7.     In the methods section, remove one paragraph from the following (1st one lies from line no.114-118 & the 2nd lies between line no. 120-123). They give the same meaning.

8.     Avoid using the personal pronounces such as “we, our, … etc.)

9.     Modify the following sub-titles (2.1.1. Experimental Materials and Methods) to be “Materials and Experimental Methods”

10.  The size of almost all the figures should be larger for readability. It is a challenge to read a printed hard copy.

11.  In page 7 and 8, the number of “Fig.4” is repeated two times, please adjust the numbering order.

12.  In page 13 (3.4.3. Comparative Experiment), state the reasons behind choosing those models for comparisons with the introduced model in this work.

Summery:

The manuscript is well-crafted but requires significant revisions to enhance clarity and readability. The decision to publish will be made after addressing the provided feedback.

Comments on the Quality of English Language

The manuscript is well-crafted but requires significant revisions to enhance clarity and readability.

Reviewer 2 Report

Comments and Suggestions for Authors

The manuscript presents an interesting topic on optimizing target detection of diamond nanostructures in boron-doped diamond thin films using CV and DWS-YOLOv8 model. The authors effectively discuss the challenges faced during target detection, such as noise, unclear boundaries, and mutual occlusion. They also provide a detailed explanation of their methodology, including the integration of deformable convolutional C2f module, shuffling attention mechanism, and Wise-IoU v3. The results demonstrate some improvements in detection accuracy with reduced computational complexity compared to the original YOLOv8 model.

However, there are a few areas that require clarification or expansion:

    1. The manuscript mentions fine-tuning the YOLOv8 model but does not specify which version of YOLOv8 is used (e.g., YOLOv8s). 

    2. In Table 4, it would be valuable to include the F1-score as well.

    3. Double check the typo in the MS, e.g line 344.

    4. To enhance understanding, the authors should provide a schematic workflow diagram illustrating the training process of DWS-YOLOv8. Additionally, including a pseudo code representation would be beneficial for readers to grasp the algorithm more easily.

    5. The manuscript mentions using a bounding box regression loss (Wise-IoU v3), but it would be helpful to plot the loss function evolves over epochs. 

Author Response

Comments 1: The manuscript mentions fine-tuning the YOLOv8 model but does not specify which version of YOLOv8 is used (e.g., YOLOv8s).

Response 1: Thanks very much for your professional comments! The version of YOLOv8 used in this study is YOLOv8n, as stated in the revised manuscript. The revision is highlighted in lines 18, 19, 23, 108, 117, and 396 in the revised manuscript. Thanks again!

Comments 2: In Table 4, it would be valuable to include the F1-score as well.

Response 2: Thanks a lot for the constructive comments! In the revised manuscript, the F1 scores for the different versions of YOLOv8 are added to Table 5 at line 455. In the evaluation metrics for target detection, the F1-score is the reconciled average of Precision and Recall. The F1-score takes values in the range [0, 1]. The higher values the F1-score takes, the better balance between precision and recall the model strikes. Thank you again for your valuable suggestions to improve the quality of our manuscript.

Comments 3: Double check the typo in the MS, e.g line 344.

Response 3: Thanks very much for pointing our mistakes! In the revised manuscript, all the typo mistakes have been carefully corrected and highlighted.

Comments 4: To enhance understanding, the authors should provide a schematic workflow diagram illustrating the training process of DWS-YOLOv8. Additionally, including a pseudo code representation would be beneficial for readers to grasp the algorithm more easily.

Response 4: Thanks a lot for your constructive comments! A workflow schematic that briefly demonstrates the training process of DWS-YOLOv8n is inserted in line 327. The training process of YOLOv8n includes data preparation, model initialization, forward propagation, loss computation, backpropagation, optimizer update, training iteration, model validation, model saving and test evaluation. Among, data preparation includes the collecting and labeling the training dataset. Model initialization helps to initialize the network structure and parameters of the YOLOv8 model. Forward propagation is used for feature extraction and prediction, while loss computation is achieved by a loss function to compute the error between the model predictions and the true labels. During the training process, gradients are computed by the back-propagation algorithm and the model parameters are updated according to the gradients. Then, the validation dataset is used to evaluate the model performance and monitor the overfitting phenomenon. When the training is completed, the best model parameters are saved for the application of the target detection. Finally, the performance of the training model is evaluated on the test dataset by calculating the evaluation metrics such as precision, recall, F1-score, and mAP. The overall structure and readability of the revised manuscript would be greatly optimized by the addition of flowchart. Thanks again!

Figure 1. Training process of DWS-YOLOv8.

Comments 5: The manuscript mentions using a bounding box regression loss (Wise-IoU v3), but it would be helpful to plot the loss function evolves over epochs.

Response 5: Thanks very much for constructive comments! The dependance of training loss and validation loss on the number of training rounds is plotted for DWS-YOLOv8 and YOLOv8n, respectively. As shown in Figure 11 in line 430. It is evident that the mean values of each loss function significantly decrease with the increasing iterations. The average value of the DWS-YOLOv8 loss function is significantly lower than that of the YOLOv8n, which indicates that the optimized loss function helps to speed up the convergence and improve the prediction accuracy.

Round 2

Reviewer 1 Report

Comments and Suggestions for Authors

Thank you for considering the suggestions for improving the manuscript. Substantial enhancements have been made in the revised version.